## RESEARCH ARTICLE

# Zebrafish model reveals developmental and hematopoietic functions of ADAMTS13

Samuele Sartori[1],*, Ignacio Babiloni Chust[1],*, Marco Varinelli[2], Alessandro Mattè[1], Piera Trionfini[2], Susanna Tomasoni[2] and Lucia Poggi[1],‡

## ABSTRACT

ADAMTS13 is a metalloprotease that cleaves the von Willebrand factor and prevents pathological thrombosis. Severe genetic deficiency of ADAMTS13 causes congenital thrombotic thrombocytopenic purpura, a life-threatening thrombotic microangiopathy. Increasing evidence suggests that ADAMTS13 contributes to physiological processes beyond hemostasis, including vascular development and tissue homeostasis, but these functions remain poorly understood. To address this gap, we generated a transparent, multitransgenic *adamts13^{i5}* zebrafish model and began investigating the developmental and disease-related roles of ADAMTS13 *in vivo*.

The *adamts13^{i5}* mutants recapitulated hallmark features of congenital thrombotic thrombocytopenic purpura, including erythrocyte fragmentation and schistocyte formation in adults. In larvae, ADAMTS13 loss unveiled a prothrombotic response to vascular injury, a phenotype masked in patients by thrombocytopenia. Mechanistically, ADAMTS13 deficiency impaired developmental vascular patterning, suppressed *vegfa* expression, and reduced macrophage number, accompanied by diminished inflammatory and pro-angiogenic signaling. ADAMTS13 loss disrupted hematopoietic homeostasis in adulthood, with myeloid expansion and lymphoid depletion in the kidney marrow. These findings establish ADAMTS13 as a multifaceted regulator of thrombosis, vascular development, inflammation, and hematopoietic lineage specification. The *adamts13^{i5}* zebrafish provides a powerful vertebrate model for dissecting the mechanisms of thrombotic thrombocytopenic purpura pathogenesis and identifying therapeutic strategies extending beyond hemostasis.

KEY WORDS: Adamts13, Angiogenesis, Zebrafish, Hematopoiesis, Congenital thrombotic thrombocytopenic purpura

## INTRODUCTION

ADAMTS13 (a disintegrin and metalloprotease with thrombospondin motifs 13) is a Zn-dependent metalloprotease mainly produced by hepatic stellate cells, with additional expression in endothelial and hematopoietic cells (Dong et al., 2002;

Turner et al., 2009). Circulating ADAMTS13 cleaves ultra-large von Willebrand factor (vWF) multimers, preventing spontaneous platelet aggregation and microvascular thrombosis (Furlan et al, 1996; Tsai, 1996; Zheng, 2015; Falter et al., 2023; South and Lane, 2018). Severe ADAMTS13 deficiency causes congenital thrombotic thrombocytopenic purpura (cTTP, or Upshaw-Schulman syndrome), a rare but life-threatening thrombotic microangiopathy (Sadler, 2008; Mingot Castellano et al., 2022).

Beyond hemostasis, ADAMTS13 has emerging roles in vascular biology. Studies suggest that it modulates endothelial activation, leukocyte recruitment, and inflammation, and may influence angiogenesis through vascular endothelial growth factor (VEGF) signaling and endothelial gene regulation, with relevance to placental development, diabetic retinopathy, and tissue repair (Gandhi et al., 2012; Lee et al., 2012; 2015; Feng et al., 2016; Gragnano et al., 2017; Randi, 2017; Xiao et al., 2017; Bitsadze et al., 2021; Mingot Castellano et al., 2022; Woods et al., 2023; Frimat et al., 2024). However, most evidence is from *in vitro* or *ex vivo* studies, and the *in vivo* role of ADAMTS13 in vascular patterning and angiogenesis remains poorly defined.

Vascular networks are intimately linked to hematopoiesis, particularly during development, where endothelial niches guide the emergence of hematopoietic stem and progenitor cells (HSPCs) and differentiation (Morrison and Scadden, 2014). This suggests that ADAMTS13 could influence blood cell development via endothelial or niche-mediated mechanisms, yet its role in hematopoiesis is largely unexplored.

Previous work in zebrafish showed that ADAMTS13 deficiency reproduces cTTP hallmarks, including thrombocytopenia, erythrocyte fragmentation, and spontaneous bleeding (Zheng et al., 2020). To investigate its developmental and disease-related roles, we generated a transparent, double-transgenic *adamts13* mutant zebrafish, enabling real-time imaging of vascular and hematopoietic development. This model shows that ADAMTS13 is essential for vascular patterning, macrophage differentiation, and pro-inflammatory/pro-angiogenic signaling during embryogenesis. The model also allows direct visualization of thrombus formation in larvae, uncovering a prothrombotic phenotype usually masked by thrombocytopenia in patients. In adults, ADAMTS13 deficiency disrupts hematopoietic homeostasis, leading to myeloid expansion and lymphoid depletion in the kidney marrow, the functional equivalent of mammalian bone marrow.

These findings establish ADAMTS13 as a multifaceted regulator of thrombosis, vascular development, inflammation, and hematopoietic lineage specification. The zebrafish *adamts13^{i5}* provides a powerful vertebrate platform to dissect the mechanisms of thrombotic microangiopathies and evaluate therapeutic strategies extending beyond classical hemostatic interventions.

[1]Department of Cellular, Computational and Integrative Biology (CIBIO), 38123 University of Trento, Italy. [2]Istituto di Ricerche Farmacologiche Mario Negri IRCCS, 24126 Bergamo, Italy.
*Co-first authors

‡Author for correspondence (lucia.poggi@unitn.it)

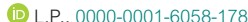 L.P., 0000-0001-6058-1781

Biology Open

## RESULTS

### Hematological phenotype in adults and validation of the *adamts13^{i5}* model

To investigate the developmental consequences of *adamts13* deficiency, we generated an *adamts13* loss-of-function allele (see Materials and Methods and Fig. S1) using a previously described CRISPR/Cas9-based strategy (Zheng et al., 2020). The resulting *adamts13^{i5}* mutant allele (hereby referred to as *a13^{i5}*) contains a 5 base pair insertion that induces a frameshift and a premature stop codon in the region encoding the signal peptide, consistent with a

predicted complete loss-of-function (Fig. S1A). To facilitate rapid and reliable genotyping, we developed an agarose gel-based assay using allele-specific oligonucleotides (ASOs) to detect the 5 bp insertion in *a13^{i5}* fish (Fig. S1B).

Flow cytometry analysis on peripheral blood from adult *a13^{i5}* mutants revealed a shift toward smaller circulating cells, indicative of schistocytosis, the presence of fragmented erythrocytes with abnormal morphology (Kunitomo et al, 2022), while cell complexity remained unchanged (Fig. 1A,B). Peripheral blood smears corroborated these findings, showing a marked reduction in circulating red blood cells

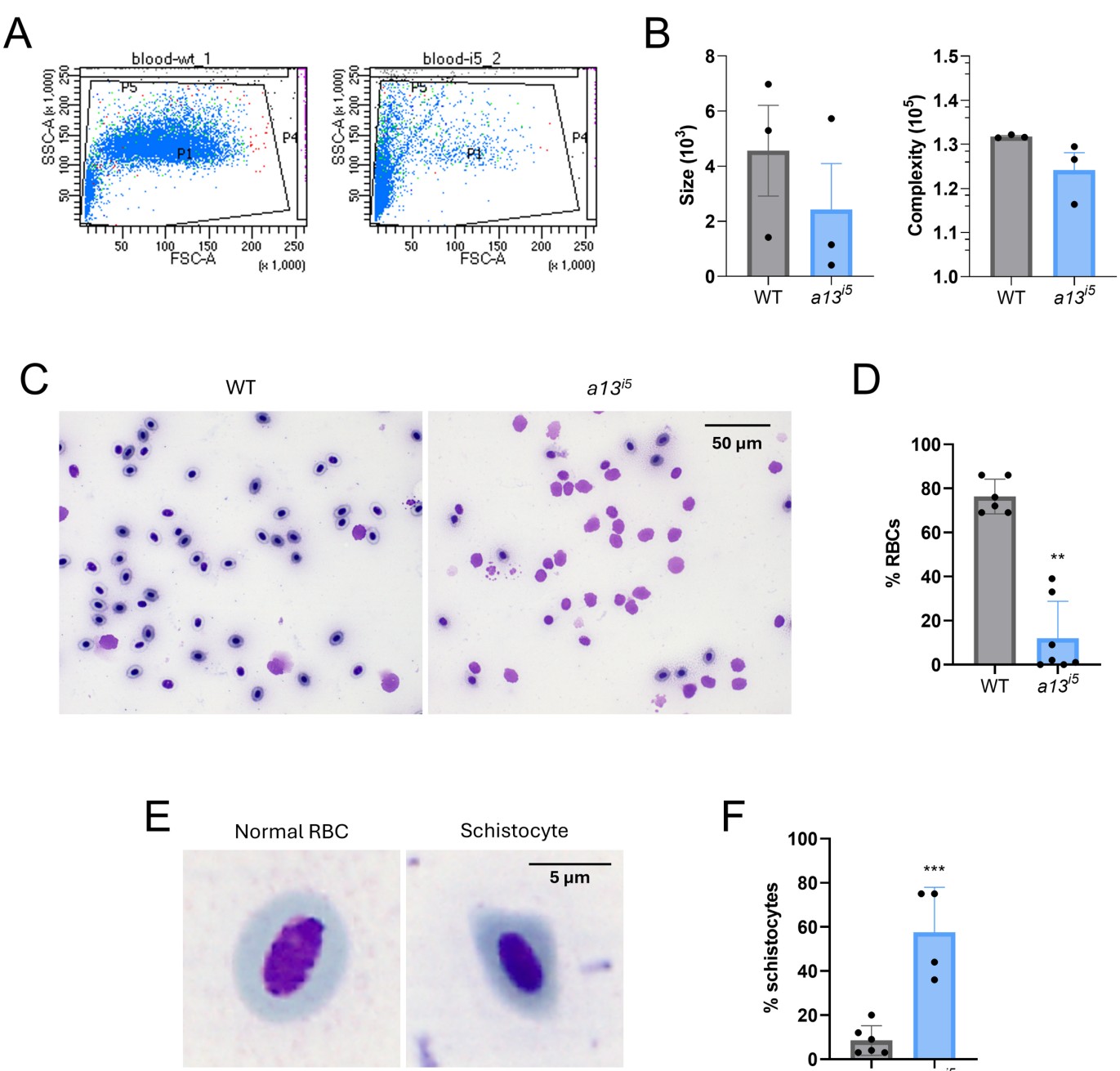

**Fig. 1. Depletion of erythrocytes and presence of schistocytes in the peripheral blood of *adamts13^{i5}* mutants.** (A) flow cytometry analysis of the peripheral blood of adult zebrafish. (B) Analysis of the size (left) and complexity (right) of WT (grey) and *a13^{i5}* (blue) fish. (C) Representative peripheral blood smears of the erythrocytes. (D) Quantification of the RBCs in the peripheral blood of adult zebrafish: WT (grey), *a13^{i5}* (blue). (E) Illustrative comparison between a normal RBC and a schistocyte. (F) Quantification of SCs in the peripheral blood of adult zebrafish: WT (grey), *a13^{i5}* (blue). Data are shown as means±s.d. of 3-7 biological replicates. Each dot represents a single adult fish. **$P \leq 0.01$ and ***$P \leq 0.001$ by two-tailed Student's *t*-test. Blood smear images were obtained using a 63× oil-immersion objective.

(RBCs) in $a13^{i5}$ mutants compared with wild-type (WT) siblings (12.0±16.8% versus 76.3±7.9%, respectively; Fig. 1C,D), consistent with the hemolytic features characteristic of cTTP and previously reported in zebrafish (Zheng et al., 2020). A hallmark of cTTP is the presence of schistocytes. Consistent with this, $a13^{i5}$ mutants exhibited a significant increase in schistocytes (Fig. 1E,F). Collectively, these results validate the $a13^{i5}$ line as a robust zebrafish model that recapitulates core hematological features of cTTP, consistent with previous mammalian and zebrafish studies (Motto, 2005; Zheng et al., 2020).

### Developmental expression of *adamts13* and early thrombotic phenotype of *a13^{i5}*

While ADAMTS13's role in adult hemostasis is well established (Zheng et al., 2020, 2022), its spatiotemporal expression and functional requirements during development have not been systematically characterized. Whole-mount *in situ* hybridization (WISH) revealed strong *adamts13* expression in the developing liver primordium at 3 days post fertilization (dpf), consistent with the liver

being the primary site of ADAMTS13 production in mammals (Uemura et al., 2005; Zhou et al., 2005; Fig. 2A). Interestingly, additional expression is detected in the anterior ocular segment, initially localized to the presumptive lens and corneal epithelium, and then restricted to the outermost periocular regions at 5 dpf, consistent with the lens epithelium, cornea, and hyaloid vasculature.

To investigate the developmental pathophysiology of *adamts13* loss-of-function *in vivo*, we generated a multi-transgenic $a13^{i5}$ line by sequentially crossing the mutant with reporter fish for erythrocytes (*gata1:dsRed*) and vasculature (*kdrl:eGFP*) and then introducing it into the transparent *casper* background (Fig. 2B,C; see Materials and Methods for details on fish line generation).

In cTTP, microvascular thrombosis is a hallmark feature, although it is often difficult to detect directly; instead, bleeding symptoms caused by thrombocytopenia are the most clinically evident manifestations (Sadler, 2008; Zheng, 2015). To assess thrombogenic propensity *in vivo*, we performed tail transection assays 5 dpf. Live imaging revealed a progressive accumulation of erythrocytes at the transection site within minutes after injury, while the heartbeat remained clearly

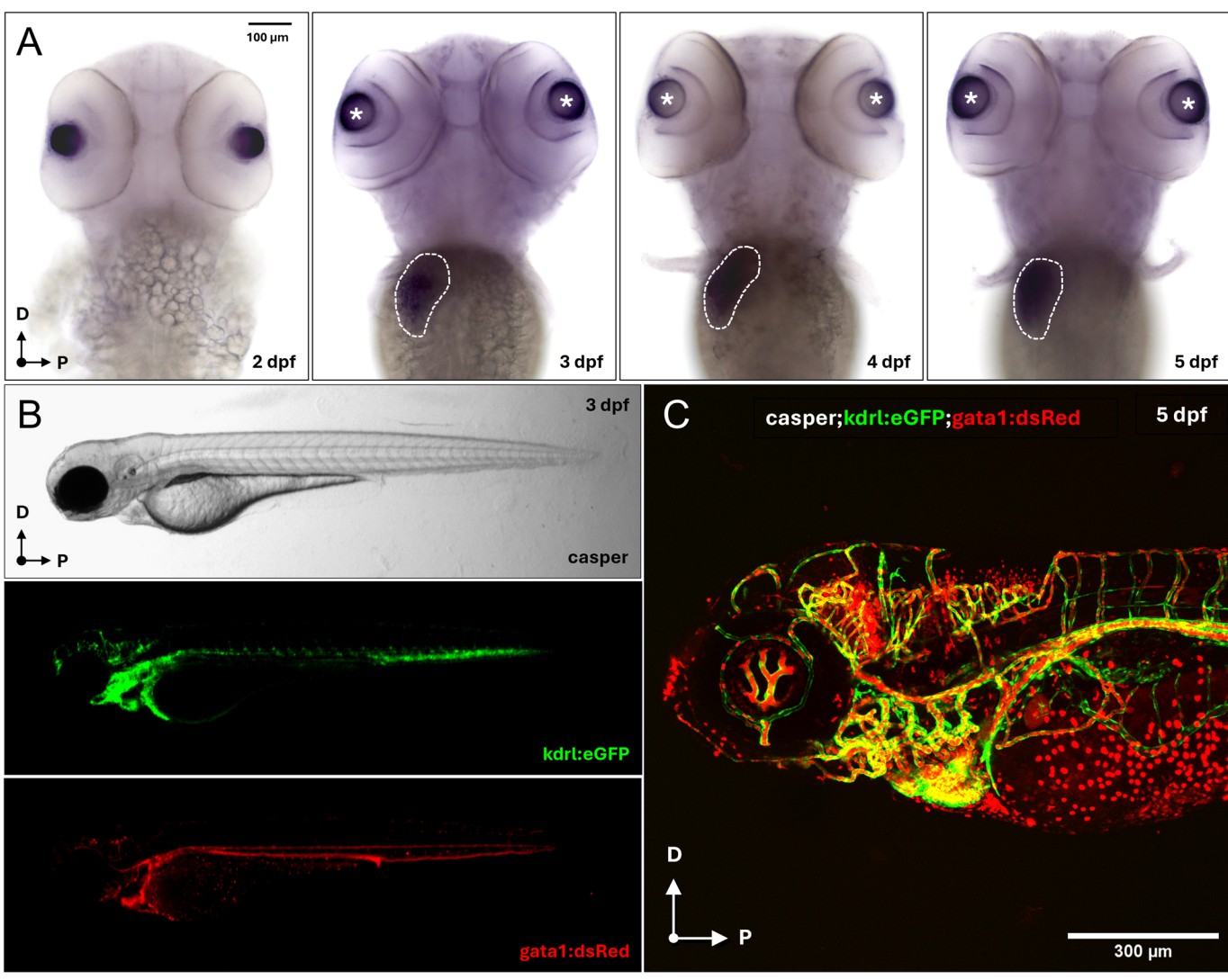

**Fig. 2. *adamts13* expression in the zebrafish liver primordium.** (A) Dorsal view of 2, 3, 4 and 5 dpf WT embryos stained through WISH using a DIG-labeled antisense RNA probe against *adamts13* (anterior to the top). Specific expression was detected at the level of the liver primordium (white dashed circle) starting from 3 dpf. Magnification: 10×. Scale bar: 100 μm. (B) Lateral view of a 3 dpf *casper;kdrl:eGFP;gata1:dsRed* embryo at the stereomicroscope (anterior to the left), labeling the blood vessels (green) and the erythrocytes (red). (C) Lateral view of a 5 dpf *casper;kdrl:eGFP;gata1:dsRed* embryo at the confocal microscope with the two transgenes merged together. 20× dry objective. Scale bar: 300 μm.

Biology Open

visible (Movie 1). The $a13^{i5}$ larvae exhibited significantly reduced occlusion times (10±3 min) compared to wild-type controls (20 ±2 min), indicative of a hypercoagulable state (Fig. 3A,B). Therefore, Adamts13 requirements for proper blood hemostasis likely arise early at the larval stage and can be effectively visualized in real time, reinforcing the utility of the zebrafish model for studying early thrombotic manifestations of ADAMTS13 deficiency.

### *Adamts13* loss-of-function impairs vascular development and downregulates *vegf* expression

Emerging evidence suggests that ADAMTS13 participates in angiogenic processes beyond its role in VWF cleavage and hemostasis (Lee et al., 2012, 2015; Abu El-Asrar et al., 2025). To assess its contribution to vascular development, we analyzed the effect of the $a13^{i5}$ mutation on the trunk vasculature, which has a stereotyped and highly organized pattern. During vascular development, one intersegmental vessel (ISV) sprouts from the dorsal aorta (DA), runs between each pair of somites, and connects to the dorsal longitudinal anastomotic vessel (DLAV). Venous ISVs arise instead from the posterior cardinal vein (PCV) to finally close the loop (Bower et al., 2017; Eberlein et al., 2021). By 5 dpf, the ISVs in the trunk of zebrafish embryos are well developed and differentiated into arteries and veins, forming a functional vascular network (Wacker et al., 2014; Hogan and Schulte-Merker, 2017). At this stage, $a13^{i5}$ mutants exhibited significantly shorter ISVs (16% normalized to DAPI staining) compared to wild-type controls (Fig. 4A,B), with 68% of them displaying aberrant anastomosis (Fig. 4C). Additionally, the subintestinal vein (SIV) plexus was consistently more hypoplastic in mutants (Fig. 4D).

To determine whether these defects reflect a developmental delay or a sustained disruption, we examined larvae at 7 dpf. ISV length remained significantly shorter in $a13^{i5}$ mutants (21% normalized to DAPI staining) compared to wild-type controls, and aberrant vessel patterning persisted in 87% of mutants (Fig. S2A,B). The SIV plexus also remained underdeveloped in approximately 40% of $a13^{i5}$ larvae. These findings indicate a persistent impairment in angiogenic patterning rather than a transient developmental delay.

One proposed mechanism by which ADAMTS13 regulates vascular development is modulation of VEGF signaling (Lee et al., 2012, 2015). To gain mechanistic insights, we analyzed *vegfa*

expression, which was significantly downregulated by 75% in $a13^{i5}$ mutants (Fig. 4E).

Together, these data suggest that an interplay between ADAMTS13 and VEGF-mediated angiogenesis is required for vascular development, including ISV elongation, dorsal vessel connectivity, and subintestinal plexus formation, potentially through upregulation of *vegfa* by ADAMTS13. The zebrafish larva, with its stereotyped vasculature and transparent body plan, in combination with the $a13^{i5}$ mutant and transgenic fluorescent reporter lines, offers a tractable model to investigate the role of ADAMTS13 in vascular development and remodeling.

### The $a13^{i5}$ mutation alters the expression of pro-angiogenic factors and macrophage markers

Angiogenesis is tightly coupled to inflammatory signaling, with pro-inflammatory cytokines such as IL-1β, IL-6, and TNFα acting as potent mediators of vascular remodeling (Fahey and Doyle, 2019; Jeong et al., 2021; Kelly and Panigrahy, 2023). Within this context of the aberrant angiogenesis and due to the emerging link between ADAMTS13 and inflammation (Lu et al., 2020; Abu El-Asrar et al., 2025), we quantified cytokine expression in $a13^{i5}$ larvae at 5 dpf. While *il-1β* levels were variable and not significantly altered (Fig. 5A), both *il-6* and *tnfα* were significantly downregulated in mutants, by 70% and 50%, respectively (Fig. 5B,C).

As these cytokines are predominantly secreted by macrophages (Nguyen-Chi et al., 2015; Tsarouchas et al., 2018), which are also key regulators of angiogenesis through the secretion of VEGF, and other mediators (Granata et al., 2010; Graney et al., 2020), we next examined macrophage-specific markers (*mpeg1* and *marco*). Expression of *mpeg1* was significantly reduced (~75%) in $a13^{i5}$ mutants (Fig. 5D), while *marco* exhibited a non-significant downward trend (Fig. 5E). These findings suggest a mechanism whereby impaired macrophage specification, differentiation, or recruitment may contribute to angiogenic defects as observed in the absence of ADAMTS13.

### ADAMTS13 deficiency disrupts hematopoietic lineage composition in the adult kidney marrow

Given the reduced expression of macrophage-associated genes and cytokines observed in larvae and peripheral blood abnormalities in

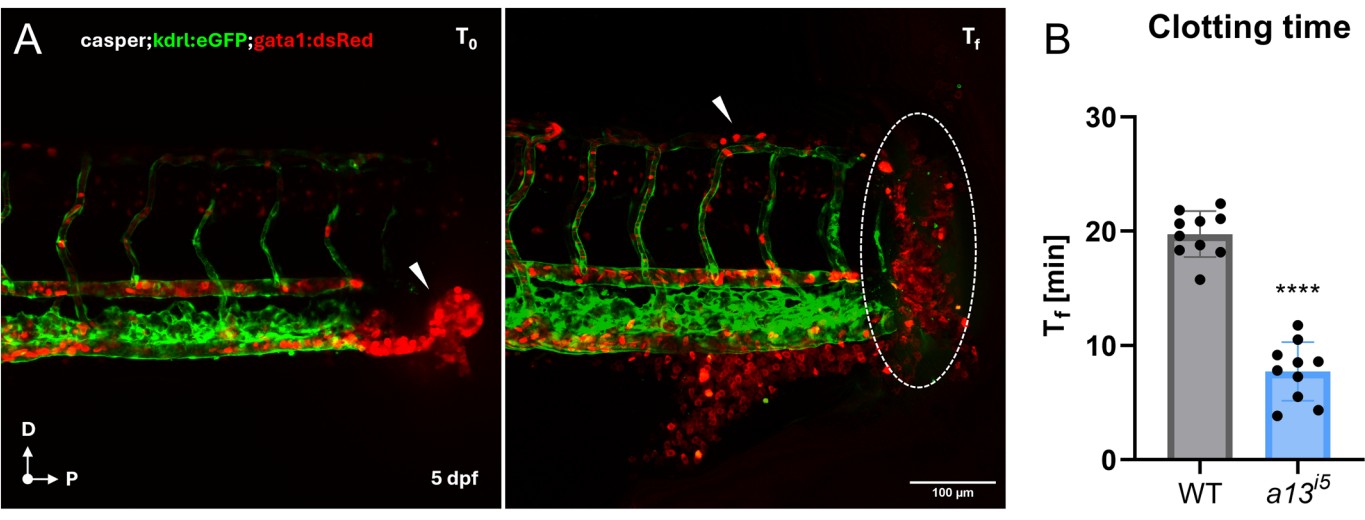

**Fig. 3. Accelerated clotting in the absence of ADAMTS13.** (A) Representative confocal images of a *casper;kdrl:eGFP;gata1:dsRed* zebrafish at 5 dpf at time 0 ($T_0$) and at the final clotting time ($T_f$). (B) Quantification of the clotting time ($T_f$ in minutes) in WT (grey) and $a13^{i5}$ (blue) zebrafish larvae. Data are shown as means±s.d. Each dot represents a single embryo. ****$P \leq 0.0001$ by two-tailed Student's *t*-test. Scale bar: 100 µm.

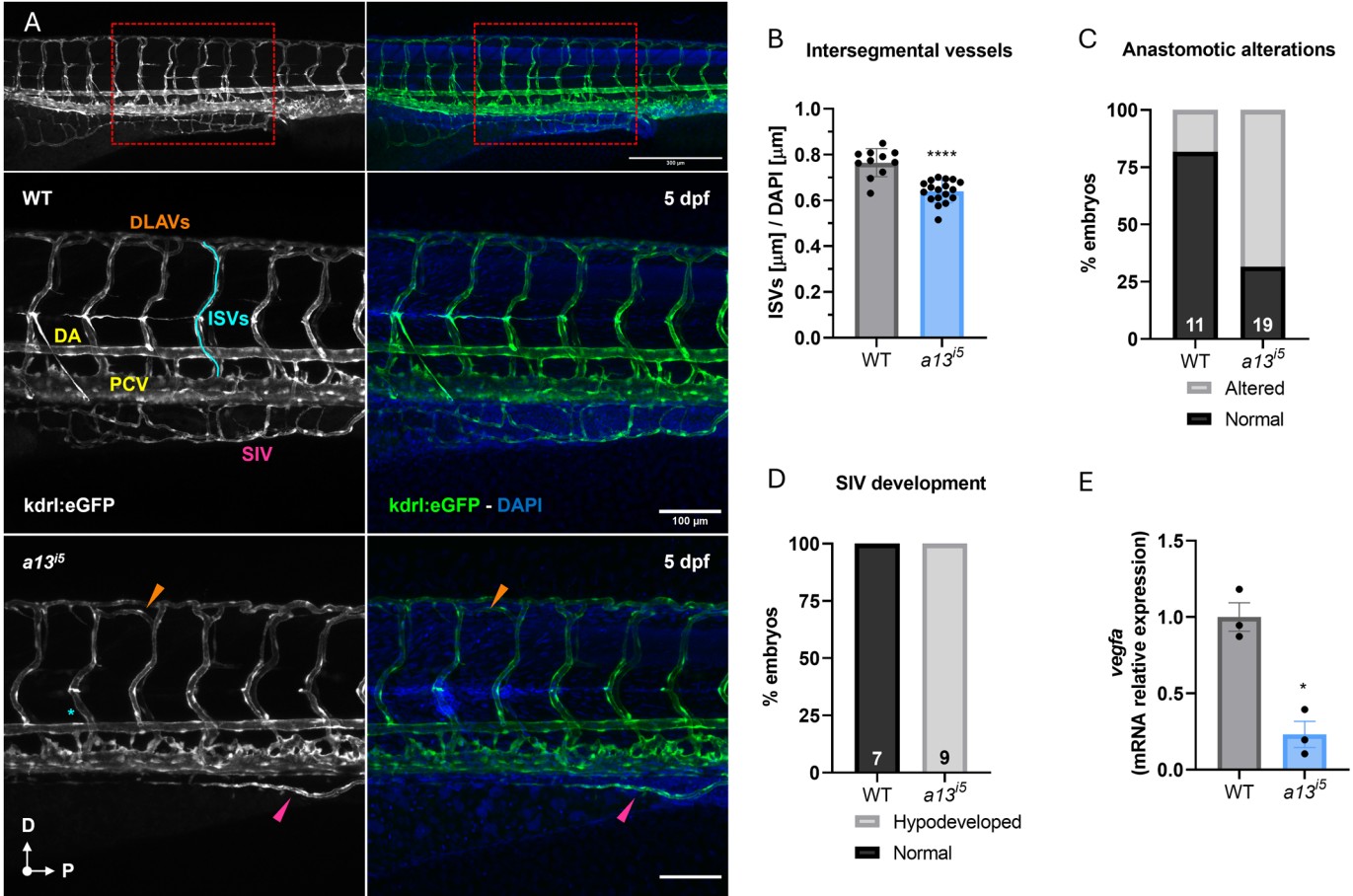

**Fig. 4. ADAMTS13 deficiency impairs vascular developmental patterning.** (A) Representative confocal images of the trunk vasculature of *casper;kdrl: eGFP* transgenic zebrafish at 5 dpf. Top panels show an overview of the tail vasculature (10×), the red rectangle highlights the area of the trunk considered for the analysis. Scale bar: 300 μm. Mid panels show a WT embryo, *kdrl:eGFP* in grey (left), and the merge with DAPI (right). Lower panels show an *a13^i5* mutant embryo, *kdrl:eGFP* in grey (left) and the merge with DAPI (right). Scale bar: 100 μm. (B) Quantification of ISV length (μm) normalized to DAPI in WT (grey) and *a13^i5* (blue) embryos. (C) Percentage of embryos with normal (black) or altered (grey) anastomotic connections in WT and *a13^i5* mutants. (D) Percentage of embryos with normal (black) or hypodeveloped (grey) SIV plexus in WT and *a13^i5* mutants. (E) Relative mRNA expression levels of *vegfa* in WT and *a13^i5* embryos, normalized to the housekeeping gene *ube2a*, and shown as fold versus WT. Data are shown as mean±s.d. Each dot represents a biological replicate. *$P \leq 0.05$, **** $P \leq 0.0001$ by unpaired Student's *t*-test. DA, dorsal aorta; DLAV, dorsal longitudinal anastomotic vessel; ISV, intersegmental vessel; PCV, posterior cardinal vein; SIV, subintestinal vein.

adults, we hypothesized that *adamts13* loss-of-function may also affect adult hematopoiesis. In zebrafish, the kidney marrow (KM) is the primary hematopoietic organ during adulthood (LeBlanc et al., 2007; Mahony and Monteiro, 2024). Flow cytometry of KM cell suspensions from wild-type and *a13^i5* adults revealed significant alterations in lineage composition. Myeloid cells were markedly expanded in mutants (Fig. 6A,B), whereas lympho-erythroid cells were significantly reduced (Fig. 6A,C). Progenitor cell frequencies were not significantly affected (Fig. 6A,D).

RT-qPCR analysis of lineage-specific transcripts corroborated these findings. Expression of *mpeg1* and *marco*, markers of macrophage differentiation, was significantly decreased in *a13^i5* KM (Fig. 6E,F), as well as *gata1*, a key erythroid transcription factor (Aluri et al., 2025) and *mpx*, a marker of neutrophil differentiation (García-López et al., 2023; Fig. 6G,H). These findings suggest that ADAMTS13 is required for hematopoietic lineage maturation in the KM. The *a13^i5* zebrafish model thus provides a valuable *in vivo* platform for investigating the broader physiological functions of ADAMTS13 beyond its canonical role in hemostasis, including identifying inflammatory and angiogenic signaling whereby this function is regulated.

## DISCUSSION

Previous zebrafish models of ADAMTS13 deficiency studies primarily focused on recapitulating adult cTTP features such as schistocytosis, prolonged bleeding, and prothrombotic phenotypes in *ex vivo* assays (Zheng et al., 2020). Our newly generated *adamts13^i5* line reproduces these hallmarks and enables real-time, *in vivo* investigation of thrombotic microangiopathy across developmental stages.

Most prior studies of ADAMTS13 in animal models have focused on its role in hemostasis and TTP, or on its anti-inflammatory effects under stress or pathological conditions (Abu El-Asrar et al., 2025; Matsui et al., 2014; Xiao et al., 2017; Zheng et al., 2020), with little exploration of developmental or basal immune functions. Thus, while ADAMTS13 is clearly essential for thrombotic regulation, its broader roles outside hemostasis remain largely unexplored. Our study provides new insights by revealing developmental functions of ADAMTS13 in angiogenesis, immune regulation, and hematopoiesis under baseline conditions.

Crossing the mutant line into the transparent *casper* background and introducing *gata1:dsRed* and *kdrl:eGFP* reporters, we established a robust *in vivo* platform for visualizing microvascular

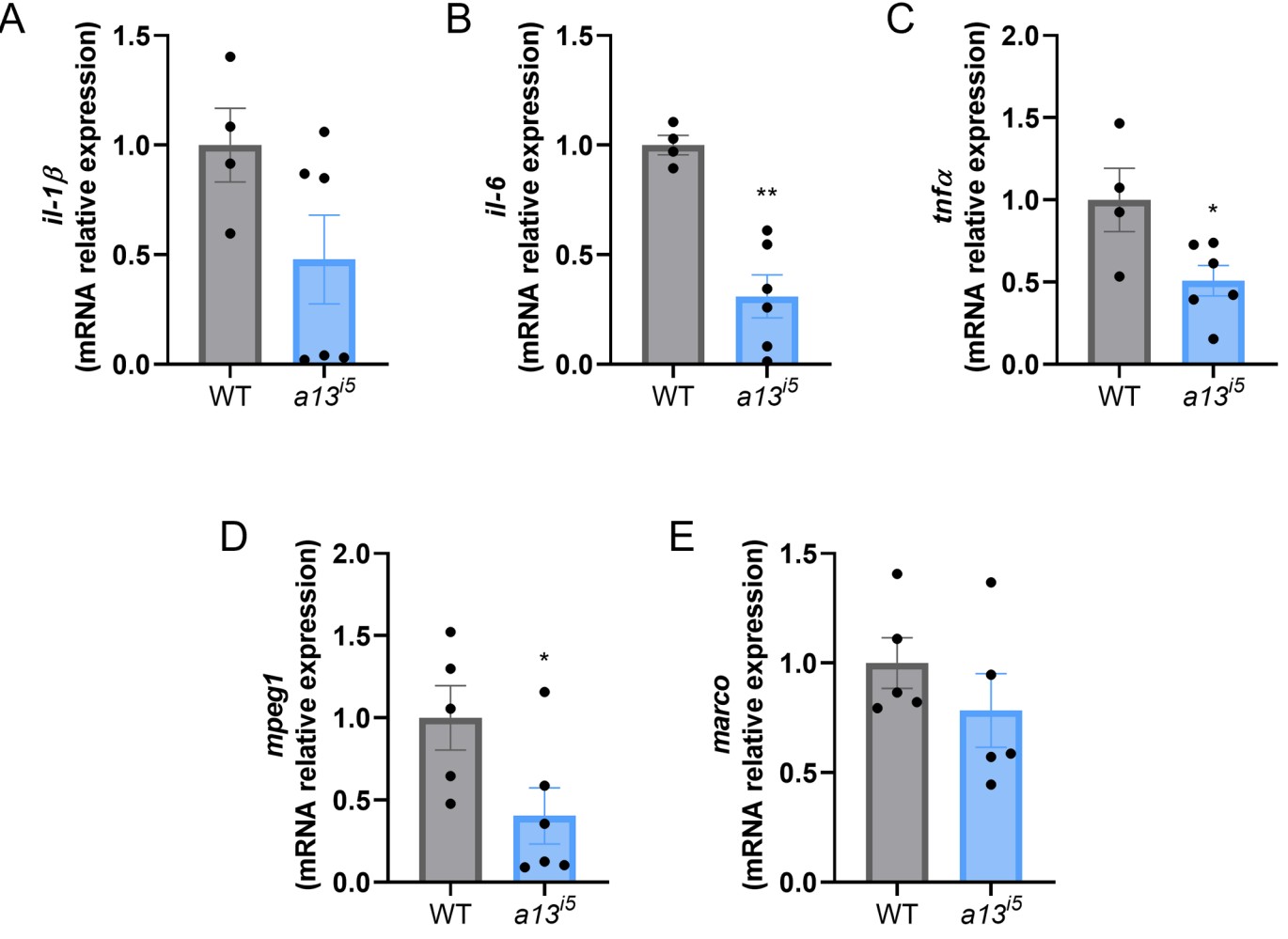

**Fig. 5. ADAMTS13 regulates inflammation.** mRNA expression of the pro-inflammatory cytokines *il-1β* (A), *il-6* (B), *tnfα* (C) and the macrophage-specific markers *mpeg1* (D) and *marco* (E) in WT (grey) and *a13$^{i5}$* mutants (blue) at 5 dpf. mRNA expression was measured by RT-qPCR, normalized to the housekeeping gene *ube2a*, and shown as fold versus WT. Each dot represents a pull of 20 embryos. Data are shown as mean±s.d. of four independent experiments. *$P \leq 0.05$, **$P \leq 0.01$ and ***$P \leq 0.001$ by two-tailed Student's *t*-test.

thrombosis and dissecting underlying mechanisms. Using this approach, we observed an early onset prothrombotic phenotype in larvae, with accelerated clot formation after vascular injury. This provides direct evidence of the typical thrombotic microangiopathy of cTTP and access for *in vivo* studies. The conserved hepatic expression of *adamts13* in zebrafish larvae, mirroring mammalian ADAMTS13 (Zhou et al., 2005), further supports the translational relevance of this model. Collectively, the *adamts13$^{i5}$* zebrafish offers a unique tool for dissecting early cTTP mechanisms and for high-throughput pharmacological screening targeting ADAMTS13-related pathologies.

Beyond canonical hemostasis, our findings uncovered developmental roles for ADAMTS13 in vascular patterning and immune regulation. Homozygous mutants exhibit vascular anomalies, including shortened intersegmental vessels, defective anastomoses, and hypoplastic subintestinal vein plexuses, accompanied by downregulation of *vegfa*. These data support a conserved role for ADAMTS13 in promoting angiogenesis, consistent with prior *in vitro* studies showing that it enhances VEGF signaling and endothelial proliferation, including retinal endothelial cells (Lee et al., 2012, 2015; Xiao et al., 2017; Woods et al., 2023; Dutta Gupta and Ta, 2024). However, the causal relationship between ADAMTS13 and VEGF

signaling and the specific cell types involved remains unresolved. Although hepatocytes are the primary source of ADAMTS13, a minor secretion from endothelial cells has been reported (Turner et al., 2009), raising the intriguing possibility of an autocrine effect. To test this directly, future experiments using cell transplantation, recombinant ADAMTS13, mRNA injection, or pharmacological VEGF agonists will be required.

Interestingly, the angiogenic defects coincide with the suppressed expression of *il-6*, *tnfα* and the macrophage marker *mpeg1*, suggesting that ADAMTS13 modulates the immune microenvironment. Because macrophages are critical regulators of angiogenesis and vascular remodeling (Martin and Gurevich, 2021; Shah and Lee, 2024), we hypothesize that reduced macrophage number or altered phenotype of these cells contributes to the observed vascular defects. While this hypothesis requires direct testing through functional experiments, such as macrophage depletion or knockdown models (Yang et al., 2020; Yang et al., 2021), these findings highlight the dual regulation of endothelial and immune pathways by ADAMTS13. Inflammation is a hallmark of acute TTP episodes (Westwood et al., 2014; Demeter et al., 2024), and inflammatory cytokines can suppress ADAMTS13 expression in hepatic and endothelial cells (Cao et al., 2008). Our data showing reduced pro-inflammatory signaling in *adamts13$^{i5}$* larvae

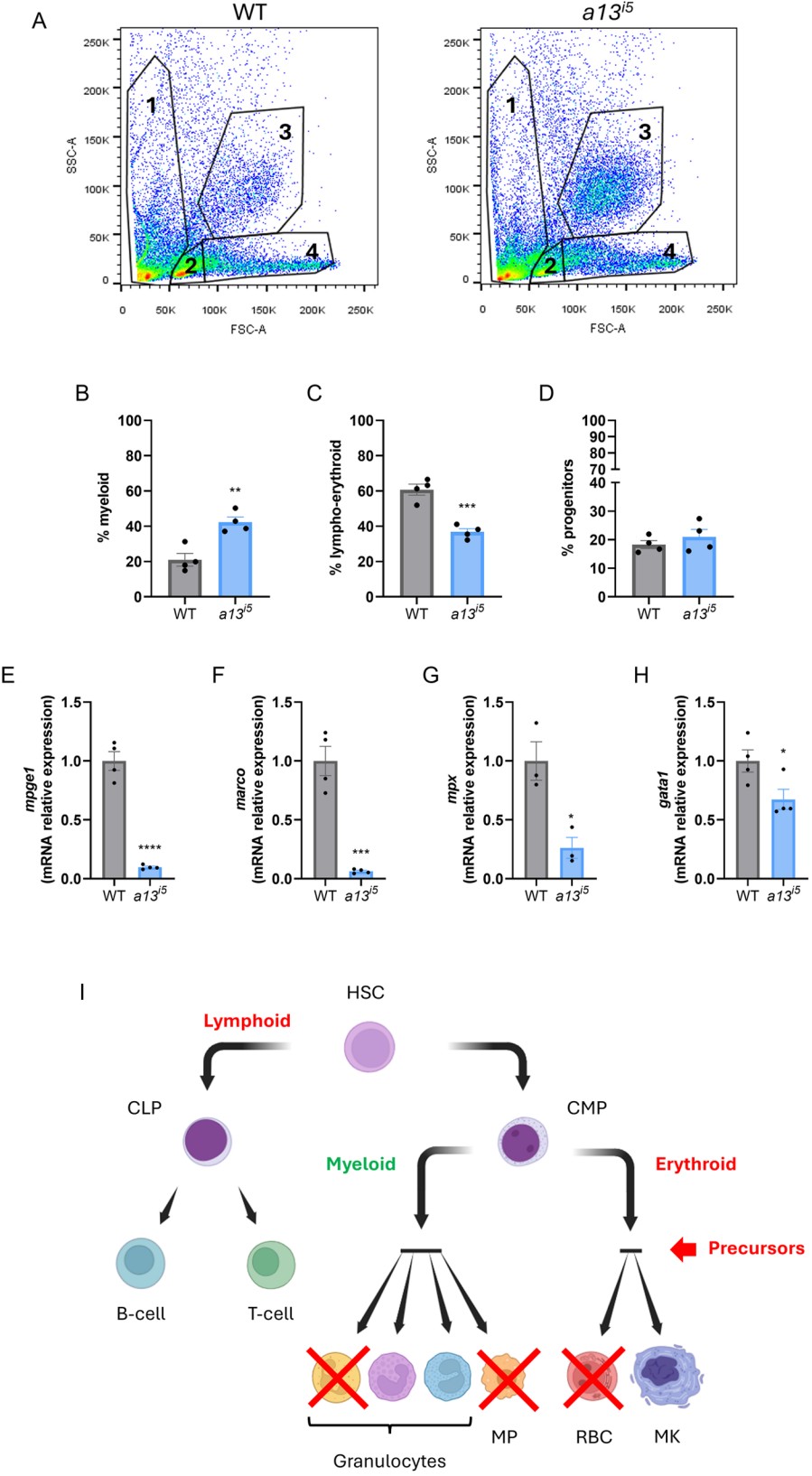

**Fig. 6. ADAMTS13 is required for proper hematopoietic lineage maturation in the adult zebrafish.** (A-D) Flow cytometry analysis of kidney marrow-derived cells from WT (grey) and *a13*$^{i5}$ (blue) adult zebrafish. (A) Gating strategy highlighting the main KM cell populations: cell debris (1), lympho-erythroid (2), myeloid (3) and early progenitors (4). Bar graphs represent the relative proportions of myeloid (B), lympho-erythroid (C), and progenitor (D) cell populations. (E-H) RT-qPCR analysis of lineage-specific gene expression in WT and *a13*$^{i5}$ mutants. Expression of the macrophage markers *mpeg1* (E) and *marco* (F), the neutrophil marker *mpx* (G) and the erythroid marker *gata1* (H) normalized to the housekeeping gene *ube2a* and shown relative to the WT. Data are represented as mean±s.d. Each dot represents a biological replicate. Statistical analysis was performed using a two-tailed Student's *t*-test: *$P \leq 0.05$, **$P \leq 0.01$, ***$P \leq 0.001$, ****$P \leq 0.0001$. (I) Summary scheme of the effect of the lack of *adamts13* on hematopoietic lineage specification. HSC, hematopoietic stem cell; CLP, common lymphoid progenitor; CMP, common myeloid progenitor; MP, macrophage; RBC, red blood cell; MK, megakaryocyte.

point to a primary developmental defect in immune cell specification or niche formation. Future rescue experiments will be needed to test this hypothesis directly.

Our data also reveal a role for ADAMTS13 in hematopoietic lineage specification in the adult. Flow cytometry of adult kidney marrow in *adamts13*$^{i5}$ mutants show myeloid expansion and lymphoid reduction, accompanied by reduced expression of *mpeg1*, *marco*, *mpx*, and *gata1*. These alterations appear developmentally rooted, as similar skewing is seen in embryos, and parallel findings in murine bone marrow transplantation models

indicate that ADAMTS13 is essential for hematopoietic niche establishment (Matsui et al., 2014). In support of translational relevance, recombinant ADAMTS13 has been shown to restore neutrophil and platelet reconstitution, and reduced activity correlates with increased infection risk and mortality in BMT patients (Peyvandi et al., 2006; Matsui et al., 2014). Notably, compensatory myeloid–erythroid expansion may reflect increased consumption of erythrocytes and platelets, as described in immune thrombocytopenia models (Herd et al., 2021; Stanley et al., 2025). Consistent with this, the prothrombotic activity and RBC fragmentation in $adamts13^{i5}$ mutants likely drive compensatory expansion. However, a limitation of our current approach is that it cannot discriminate among the multiple intermediate progenitor populations involved in hematopoietic maturation. Future studies, such as cell transplantation and single-cell analyses of kidney marrow, will clarify which specific maturation step (e.g. stem/ progenitor self-renewal, or cell fate commitment/differentiation) is disrupted.

Our findings position ADAMTS13 as a multifaceted developmental regulator influencing thrombosis, vascular morphogenesis, immune regulation, and hematopoietic balance. These data expand on prior zebrafish models and challenge the view of ADAMTS13 as primarily a hemostatic protein. Nevertheless, the lack of explicit mechanistic validation is a key limitation of the present study. Specifically, while we observe clear associations between ADAMTS13 deficiency, reduced VEGF signaling, and altered macrophage populations, these remain correlative, and we cannot yet determine whether the vascular and immune changes are primary or secondary effects. Future validation experiments involving targeted pathway modulation (e.g. VEGF receptor inhibition, macrophage depletion, or lineage-specific manipulations) will be essential to resolve this question and fully delineate the causal network regulated by ADAMTS13.

In conclusion, the $adamts13^{i5}$ zebrafish offers an unprecedented platform to investigate both developmental and pathological roles of ADAMTS13. By revealing its impact on thrombosis, angiogenesis, immune regulation, and hematopoiesis, this model establishes a foundation for mechanistic and translational research that can accelerate discovery and inform new therapeutic strategies for thrombotic microangiopathies and related disorders.

## MATERIALS AND METHODS

### Fish lines generation and maintenance

All procedures were conducted per the University of Trento ethics committee and were approved by the Italian Ministry of Health. The following zebrafish lines were used: *mitfa/mpv17 casper* (White et al., 2008), *Tg(gata1:dsRed)* (Namit et al., 2022), *Tg(kdrl:eGFP)* (Liu et al, 2019) and *TU* (Tübingen; wild type). To generate the *adamts13* knockout (KO) line (designated $a13^{i5}$), one-cell stage wild-type (WT) zebrafish embryos were microinjected intracellularly with the following mixture: 2 µl of 500 pg/nL SpCas9-GFP mRNA (Sigma-Aldrich, CAS9GFPPRO), 1 µl of 100 µM sgRNA (5′-GCCTCCCTTTGAGATAGTGT-3′), and 2 µl of MilliQ H$_2$O. At 3 months of age, genomic DNA was extracted from injected F0 fish via fin clipping. PCR amplification of the targeted locus was performed using the primers described in the Table S1.

The presence of a 5-base pair insertion allele was confirmed by Sanger sequencing (Mix2Seq Kit, Eurofins) and analyzed using the CRISPR Edits (ICE) tool (Synthego). F0 founders carrying the insertion were outcrossed to wild-type TU zebrafish to generate the F1 generation. Heterozygous F1 carriers were identified via fin-clipping, genotyped at 3 months of age, and intercrossed to produce homozygous *adamts13* mutant F2 fish. This process led to identifying a stable germline mutant carrying a 5 bp insertion, which was designated $adamts13^{utn10}$ and is hereby referred to as $a13^{i5}$.

The $a13^{i5}$ line was further crossed with the *mitfa/mpv17* (*casper*), *Tg(gata1:dsRed)* and *Tg(kdrl:eGFP)* lines through successive generations to establish a transparent, double transgenic background in which both blood cells and vasculature are fluorescently labeled.

### RNA extraction, cDNA synthesis and RT-qPCR

Total RNA was isolated from a pool of 20 embryos at 5 dpf, or from a whole adult kidney marrow using TRIzol$^{TM}$ Reagent (Thermo Fisher Scientific, 15596018) according to the manufacturer's instructions. The RNA was quantified using NanoDrop (Thermo Fisher Scientific, ND-2000C). cDNA was synthesized from 1000 ng of total RNA using an iScript$^{TM}$ cDNA Synthesis Kit (Bio-Rad, 1708891). Real-time quantitative PCR was performed in a CFX96 Real-Time System (Bio-Rad Laboratories, 3600037). The reactions were performed using an iTaq Universal SYBR Green Supermix (Bio-Rad, 1725121) in 10 µl, following the manufacturer's instructions. The values were analyzed with CFX Manager Version 1.6 (Bio–Rad) and expressed as the relative expression compared to WT values ($2^{-\Delta\Delta Ct}$). *ube2a* was used as a housekeeping gene, and all the measurements were performed in triplicate. The primers sequences are described in Table S1.

### DIG-labelled antisense RNA probe synthesis

To synthesize the antisense probe used for WISH, total RNA was extracted from 5 dpf WT embryos using TRIzol$^{TM}$ Reagent. cDNA was retrotranscribed using the SuperScript® III First-Strand Synthesis System (Invitrogen). The template cDNA was amplified starting from the total cDNA using the primers containing the T7 polymerase described in the Table S1.

The *adamts13* antisense RNA probe was generated using the T7 RNA Polymerase (Thermo Fisher Scientific, EP0111) combined with a digoxigenin RNA labeling mix (Roche, 11277073910) and then purified using a RNeasy® Mini Kit (QIAGEN, 74104). The concentration and correct size of the probe (892 bp) were checked using a 2% agarose gel and the NanoDrop system. The purified product was diluted in 100% formamide (Roth) to reach a stock concentration of 80 ng/µl (30×). The RNA probe was finally stored at −20°C or immediately diluted in hybridization mix (HybMix) for WISH.

### Whole-mount RNA *in situ* hybridization

Fixed embryos stored at −20°C were rehydrated by washing them with 75%, 50% and 25% methanol in PBST and then permeabilized with 10 µg/ml proteinase K (Sigma-Aldrich) in PBST for 15 min/dpf. After washing with PBST, the embryos were refixed for 20 min in 4% paraformaldehyde (PFA). WISH was performed following standard procedures (Thisse and Thisse, 2008) by using the DIG-labeled *adamts13* antisense RNA probe diluted in HybMix, anti-digoxigenin FAB fragments (Sigma-Aldrich) and BM Purple AP Substrate (Roche, 11442074001) for the colorimetric reaction.

### Blood smears and kidney marrow explants preparation

1-year-old fish were euthanized using 400 mg/L tricaine and placed in 0°C water to preserve blood cell integrity. Peripheral blood was extracted by excising the tail at the level of the anal fin and placing the fish in a perforated 1.5 ml tube with a 40 µm Corning® Cell Strainer filter (Sigma-Aldrich, CLS431751). The tube was then placed into another 1.5 ml tube containing the anti-coagulant solution: 5 mM EDTA (Sigma-Aldrich, 798681) in PBS supplemented with 10% fetal bovine serum (FBS, Gibco®, 10270106). Peripheral blood was collected by centrifugation at 100 $g$ for 1 min and mixed by pipetting.

Kidney marrow explants were instead prepared by cutting the abdomen of the euthanized fish. The mesonephros was extracted using a pair of forceps and homogenised in 500 µl of 10% FBS/PBS by pipetting thoroughly. The preparation was filtered with a 40 µm cell strainer. 10 µl of blood was placed on the side of a microscope slide (Thermo Fisher Scientific, 12392138) and smeared using a coverslip (DURAN). Smear staining was performed using the Histoline Bio-Diff kit (1 min fixation in solution 1, 3 min staining in solution 2, 3 min staining in solution 3 and 1 min washing in buffer solution pH 7.5). The slides were left to dry, and the coverslips were finally applied with DPX histological mounting medium (Sigma-Aldrich, 06522).

## Flow cytometry

Flow cytometric analysis was performed using 300 µl of kidney marrow extract or 30 µl of peripheral blood diluted in 270 µl of 10% FBS/PBS. The samples were filtered through a 40 µm cell strainer and transferred into 5 ml FACS tubes (pluriSelect). Flow cytometry was performed on a FACSymphony™ A1 Cell Analyzer (BD Biosciences) using forward (FSC) and side scatter (SSC) to assess cell size and internal complexity respectively. FSC and SSC voltages were adjusted based on WT. Doublets were removed using FSC-A versus FSC-H gating. All data were analyzed using FlowJo (BD Biosciences, version 11), applying consistent gating strategies across samples.

## Fish genotyping

3-month-old fish genotyping was performed by fin-clipping. gDNA extraction was performed by lysing the tissue in 50 µl of 50 mM NaOH (Roth) at 95°C for 20 min. The lysate was then cooled down in ice, and the pH was adjusted by adding 5 µl of Tris-HCl pH 8.0 (Roth). The gDNA mix was eventually vortexed and centrifuged at 13,000 $g$ to remove the cell debris. The PCR mix was prepared with 2.5 µl of 10× CoralLoad PCR buffer (Qiagen), 0.5 µl of 10 mM dNTPs, 0.5 µl of 10 µM forward (5′-CACAAATGAGGAGTCGGGCT-3′) and reverse (5′-TCAGCTGCTCG-CAACACATA-3′) primers, 0.4 µl of Taq polymerase (Qiagen) and 16.6 µl of water. The PCR reaction used 4 µl of gDNA mix ($T_m$=63°C, 30 cycles). The PCR product was then sequenced using a Mix2Seq Kit (Eurofins) and analyzed through the ICE CRISPR Analysis Tool (Synthego) to determine the genotype. Embryos deriving from F2 *adamts13$^{i5}$* fish were genotyped through 1.5% agarose gel PCR using the allele-specific oligonucleotides (ASOs) detailed in Table S1.

## Imaging

The Zeiss Axio Imager M2 microscope was used for WISH and blood smear imaging. After BM Purple staining (Roche, 11442074001, embryos at different developmental stages were mounted on specimen slides in glycerol (Sigma-Aldrich, G7893). WISH images were acquired with a 10× objective, while blood smears were imaged using a 63× oil-immersion objective. Brightness and contrast adjustments were performed using Zeiss ZEN analyzer 3.10. A Leica TCS SP8 laser scanning confocal microscope was used for tail vasculature morphology analysis and transgene visualization *in vivo* and on fixed tissue. Embryos were carefully embedded in 2% low-melting agarose (Sigma-Aldrich, A9414) in 35 mm glass-bottom dishes (Thermo Fisher Scientific, 50-305-806). Images were acquired using a 20× objective with the following acquisition parameters: resolution=1024×1024, acquisition speed=200, step size=2 µm, and z-stack size was adapted to image the entire tail thickness. Brightness and contrast adjustments were performed using Fiji-ImageJ2. The selection of transgenic embryos and clotting time analysis were carried out using a Zeiss SteREO Discovery.V8 microscope.

## Clotting time

Clotting time was measured in minutes using 5 dpf larvae. The embryos were placed in a drop of 1% low-melting agarose with 0.4% tricaine, and the injury was induced by cutting the tail with a scalpel under the stereomicroscope. The initial time ($T_0$) coincided with the moment the cut was performed, and the final time ($T_f$) was considered when the blood circulation in the tail stopped while the heart was still beating.

## Data analysis

GraphPad Prism 8.0 (GraphPad Software Inc., La Jolla, CA, USA) was used for all statistical analyses. The Shapiro–Wilk test was performed to determine the normality of the samples. Data are shown as the mean±s.d. Statistical analyses were performed using Student's *t*-test for parametric data, while the Mann–Whitney test was used for non-parametric data, as stated in the figure legends. Statistical significance was considered for *P*-value≤0.05.

## Acknowledgements

The authors would like to thank S. Robbiati, I. Mazzeo, S. Longhi, and M. Cont from the Model Organism Facility; R. Bertorelli and V. De Sanctis from the Next Generation Sequencing Facility; G. Scarduelli and M. Roccuzzo from the Advanced Imaging Facility; and the Cell Analysis and Separation Facility for their technical assistance during this study. Department CIBIO Core Facilities are supported by the European Regional Development Fund (ERDF) 2014–2020 and 2021–2027.

## Competing interests

The authors declare no competing or financial interests.

## Author contributions

Conceptualization: S.S., I.B.-C., M.V., P.T., S.T., L.P.; Data curation: S.S., I.B.-C., A.M.; Formal analysis: S.S., I.B.-C., A.M.; Funding acquisition: S.T., L.P.; Investigation: S.S., I.B.-C., L.P.; Methodology: S.S., I.B.-C., M.V., P.T., L.P.; Project administration: L.P.; Resources: M.V., S.T., L.P.; Software: S.S., I.B.-C., A.M.; Supervision: L.P.; Validation: S.S., I.B.-C., A.M.; Visualization: S.S., I.B.-C.; Writing – original draft: S.S., I.B.-C., L.P.; Writing – review & editing: S.S., I.B.-C., M.V., A.M., P.T., S.T., L.P.

## Funding

This study was supported by internal funding from the University of Trento to L.P., Fondazione Telethon, Italy (grant number GGP20073) to S.T., and Fondazione Aiuti Ricerca Malattie Rare A.R.M.R. to S.T. Open Access funding provided by University of Trento. Deposited in PMC for immediate release.

## Data and resource availability

All relevant data supporting the findings of this study, including graphs, flow cytometry plots, quantifications, representative images, and the *adamts13$^{i5}$* mutant line, are included in the main text, supplementary information, or publicly available through ZFIN (https://zfin.org/ZDB-ALT-251002-2). Additional datasets or specific raw data supporting the conclusions of this study are available from the corresponding author upon reasonable request.

## Peer review history

The peer review history is available online at https://journals.biologists.com/bio/lookup/doi/10.1242/bio.062265.reviewer-comments.pdf

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
