## [Peer Review File · Biology Open]

Zebrafish model reveals developmental and hematopoietic functions of ADAMTS13

Samuele Sartori, Ignacio Babiloni-Chust, Marco Varinelli, Alessandro Mattè, Piera Trionfini, Susanna Tomasoni and Lucia Poggi

DOI: 10.1242/bio.062265

Editor: Tristan Rodríguez

Review timeline

Original submission:	15 September 2025
Editorial decision:	22 September 2025
First revision received:	29 September 2025
Accepted:	30 September 2025

Original submission

First decision letter

MS ID#: bio.062265

MS Title: Zebrafish model reveals developmental and hematopoietic functions of ADAMTS13

Authors: Samuele Sartori, Ignacio Babiloni-Chust, Marco Varinelli, Alessandro Mattè, Piera Trionfini, Susanna Tomasoni and Lucia Poggi

I have now reached a decision on the above manuscript.

The reviewer reports are shown at the bottom of this email or can be accessed, together with a copy of this decision letter, by going to:

As you will see, the reviewers gave favourable reports, but raised some critical points that will require amendments to your manuscript. I hope that you will be able to carry these out, because we would like to be able to accept your paper.

At this stage, we also ask you to ensure your manuscript complies with our formatting guidelines “ please see our manuscript preparation guidelines for details. Provided you are able to fully address the referees’ comments, we are positive about publication of your paper (we accept over 95% of revision submissions) and therefore hope you won’t mind any extra work involved in reformatting your manuscript at this point.

Please upload both a ‘clean’ version of your Word file, along with a highlighted version clearly showing where you have made changes in the revised manuscript. Please avoid using ‘Track changes’ in Word files as these are lost in PDF conversion.

I should be grateful if you would also provide a point-by-point response detailing how you have dealt with the points raised by the reviewers in the ‘Response to Reviewers’ box. Please attend to all of the reviewers’ comments. If you do not agree with any of their criticisms or suggestions please explain clearly why this is so.

Reviewer 1

Comments for the author

Sartori et al. describe the haematological phenotype of mutants of ADAMTS13. This gene has been identified in congenital thrombotic thrombocytopenic purpura (cTTP). Here, the authors use

zebrafish mutants and propose this as a valuable model for studying cttp. The authors show that ADAMTS13 deficiency results in prolonged clotting times, vascular patterning defects characterised by reduced vegfa expression, altered cytokine signalling, and skewed haemopoiesis. These findings expand upon earlier zebrafish studies by Zheng et al. (2020, *Haematologica*) and Zheng et al. (2022, *J Thromb Haemost*), which focused primarily on thrombosis and thrombotic microangiopathy phenotypes in *adamts13*' fish, by uncovering new roles for ADAMTS13 in angiogenesis, immune regulation, and haemopoietic balance.

1. Experimental Quality

Controls:

- Each major figure includes appropriate wild-type (WT) controls.
- Tail transection experiments include side-by-side clotting time measurements in WT and mutants.

Appropriateness of methods:

- Methods are appropriate:

Statistics:

- Analyses include Shapiro-Wilk for normality, Student's t-test for parametric data, and Mann-Whitney for non-parametric. These are appropriate.
- p-values are clearly reported in figure legends.
- Sample sizes are indicated in figure legends.

2. Reproducibility

Replicates:

- Figures state biological replicates. These seem fine given the strength of the phenotypes observed.
- RT-qPCR is done in triplicate technical replicates, which is standard.

Raw data availability:

- Quantification is shown in graphs with individual data points.

Raw microscopy images are included.

- I could not find a statement on raw data availability in the review materials provided - access to full image sets or flow cytometry plots would be useful

Methodological detail:

- Methods are in detail: primer sequences, imaging conditions, flow cytometry gating, probe synthesis, and transgenic procedures are included.

3. Completeness

Conclusions supported by data:

- The link between ADAMTS13 deficiency and clotting, vascular abnormalities, reduced cytokine expression, and imbalanced haematopoiesis is convincingly demonstrated through multiple independent assays.
- The conclusion that ADAMTS13 regulates immune and angiogenic signalling is acceptable. It is correlative, as the reduction in macrophage markers and cytokines is demonstrated, but no direct mechanistic tests (such as rescue with recombinant ADAMTS13 or macrophage-specific knockdowns) have been conducted. This limitation should be acknowledged in the discussion.

Potential flaws or missing experiments

- The causal relationship between ADAMTS13 and VEGF signalling is inferred from vegfa down regulation but has not been directly tested (for example, rescue with vegfa mRNA or VEGF agonists).
- The haematopoietic lineage shift is described but not fully dissected; functional assays (e.g., transplantation) could clarify whether stem/progenitor self-renewal is affected.
- These are not flaws - just it would be reasonable to include these points in the discussion.

Acknowledgment of limitations

- The discussion highlights the translational importance and compensatory haematopoiesis but does not explicitly discuss the lack of mechanistic validation (e.g., whether VEGF or macrophage

changes are primary or secondary effects). The authors could mention these as validation experiments in the discussion.

4. Scholarship

Engagement with supporting/contradictory literature:

- Authors cite a broad range of previous zebrafish, mouse, and clinical studies linking ADAMTS13 to thrombosis, angiogenesis, and inflammation.
- Relevant negative or contradictory findings (e.g., studies suggesting ADAMTS13 is not essential outside haemostasis) could be discussed more explicitly to improve balance.

Position in the field

- The manuscript convincingly argues that this zebrafish model extends ADAMTS13 research beyond hemostasis, highlighting new findings in angiogenesis and hematopoiesis.

Overall recommendation

A good manuscript. Some changes in the text that bring up possible future experiments to validate some of the models presented would be helpful.

Reviewer 2

Comments for the author

In this ms, Lucia Poggi and colleagues report on the role of the metalloprotease ADAMTS13 in zebrafish development. Authors generated a new CRISPR/Cas9 loss of function allele of ADAMTS13 in a Casper background thus allowing in vivo phenotypic characterization with the use of fluorescent markers to label cells and tissues of interest. The paper expands the analysis of ADAMTS13's roles in adult stages to developing animals, and unravel new roles of this protein in development. The paper is subdivided into the following sections. First, authors validate the mutant allele by demonstrating the presence of schistocytosis, fragmentation of erythrocytes, in adult stages. This particular term should be defined the first time is being used. Authors then next characterize the expression of ADAMTS13 to the liver expression (as well as in the eye primordium), consistent with the reported expression of this gene in the mammalian liver. With the use of fluorescent markers to label the vascular system and the erythrocytes in developing animals, authors demonstrate a hypercoagulable state of mutant animals upon transection assays, defects in vascular development associated to reduced expression of vegfa expression, altered expression of cytokines and compromised macrophage specification. Authors should define the casper allele and macrophage markers (marco and mpeg1) the first time they mention them in the ms. Lastly, authors show defects in hematopoietic lineage specification (expansion of myeloid cells and reduction of lympho-erythroid without affecting the progenitor cell frequencies). The paper is well written, figures self-explanatory and will be of interest to the community interested in the associated disease. I have the following minor issues:

- (1) Define the technical terms the first time they are used (casper background, macrophage markers, etc)
- (2) Sentence in line46 of pg 10 has no-sense ("ADAMTS13 is required for defective hematopoietic lineage maturation")

Reviewer's Responses to Questions

Experimental quality

Does each figure have the proper controls?

If 'No', please indicate reasons in Comments for Author box below.

Reviewer #1:

- Yes

Reviewer #2:

- Yes

Were the data analyzed using appropriate statistical tests?
If 'No', please indicate reasons in Comments for Author box below.

Reviewer #1:

- Yes

Reviewer #2:

- Yes

Reproducibility

Were experiments performed using adequate number of biological replicates?
If 'No', please indicate reasons in Comments for Author box below.

Reviewer #1:

- Yes

Reviewer #2:

- Yes

Does the methods section provide sufficient detail to permit reproducibility?
If 'No', please indicate reasons in Comments for Author box below.

Reviewer #1:

- Yes

Reviewer #2:

- Yes

Completeness

Are the manuscript's conclusions supported by the data?
If 'No', please indicate reasons in Comments for Author box below.

Reviewer #1:

- Yes

Reviewer #2:

- Yes

Scholarship

Do the authors cite and discuss the merits of data that would argue for and against their conclusion?

If 'No', please indicate reasons in Comments for Author box below.

Reviewer #1:

- Yes

Reviewer #2:

- Yes

Does the manuscript title & abstract accurately reflect the contents of the manuscript, without hyperbole?

If 'No', please indicate reasons in Comments for Author box below.

Reviewer #1:

- Yes

Reviewer #2:

- Yes

First revisionAuthor response to reviewers' comments

We sincerely thank the reviewers for taking the time to review our manuscript. We understand that both reviewers recognised the relevance of the study and raised no concerns regarding the quality of the methodology or the robustness of the data. We greatly appreciate the constructive feedback and insightful suggestions, which we believe have helped us significantly improve the clarity, rigour, and overall quality of the work.

Below, we provide a detailed, point-by-point response to each critical comment. Where applicable, changes made in the revised manuscript are indicated with line numbers. Minor typographical and grammatical errors were also corrected throughout the manuscript, particularly in the Materials and Methods.

Reviewer 1 Comment / Response**1. Raw data availability:**

I could not find a statement on raw data availability in the review materials provided - access to full image sets or flow cytometry plots would be useful.

Response:

We thank the reviewer for highlighting this point. The main original data are already included in the manuscript. The new *adamts13i5* mutant line will be made publicly available through ZFIN (Zebrafish Information Network), and we will be happy to provide any additional raw data from

this study upon request. This information has been summarized in a dedicated “Data Availability” section of the revised manuscript:

Lines 451-456, Data Availability

All relevant data supporting the findings of this study, including graphs, flow cytometry plots, quantifications, representative images, and the adamts13ⁱ⁵ mutant line, are included in the main text, supplementary information, or publicly available through ZFIN (Zebrafish Information Network). Additional datasets or specific raw data supporting the conclusions of this study are available from the corresponding author upon reasonable request.

2. Conclusions supported by data:

Potential flaws or missing experiments

- *The causal relationship between ADAMTS13 and VEGF signalling is inferred from vegfa down regulation but has not been directly tested (for example, rescue with vegfa mRNA or VEGF agonists).*
- *The haematopoietic lineage shift is described but not fully dissected; functional assays (e.g., transplantation) could clarify whether stem/progenitor self-renewal is affected.*
- *These are not flaws - just it would be reasonable to include these points in the discussion.*

Acknowledgment of limitations

- *The discussion highlights the translational importance and compensatory haematopoiesis but does not explicitly discuss the lack of mechanistic validation (e.g., whether VEGF or macrophage changes are primary or secondary effects). The authors could mention these as validation experiments in the discussion.*

Response:

We fully agree with the reviewer’s comments and thank them for providing their insightful suggestions. In the revised manuscript, we have taken all of these considerations into account for the Discussion. We explicitly acknowledge that the observed effects on VEGF signalling, macrophage populations, and hematopoietic lineage specification remain correlative, warranting further functional data. This is particularly emphasised in the following parts:

Lines 255-260: However, the causal relationship between ADAMTS13 and VEGF signaling and the specific cell types involved remains unresolved. ... To test this directly, future experiments using cell transplantation, recombinant ADAMTS13, mRNA injection, or pharmacological VEGF agonists will be required.

Lines 264-274: Because macrophages are critical regulators ... we hypothesize that reduced macrophage number or altered phenotype of these cells contributes to the observed vascular defects... Our data ... point to a primary developmental defect in immune cell specification or niche formation. Future rescue experiments will be needed to test this hypothesis directly.

Lines 288-292: However, a limitation of our current approach is that it cannot discriminate among the multiple intermediate progenitor populations ... Future studies, such as cell transplantation and single-cell analyses of kidney marrow, will clarify which specific maturation step (e.g. stem/progenitor self-renewal, or cell fate commitment/differentiation) is disrupted.

Lines 294-304: Our findings position ADAMTS13 as a multifaceted developmental regulator... These data expand on prior zebrafish models and challenge the view of ADAMTS13 as primarily a hemostatic protein. Nevertheless, the lack of explicit mechanistic validation is a key limitation of the present study... Future validation experiments... will be essential to resolve this question and fully delineate the causal network regulated by ADAMTS13.

3. Scholarship

Engagement with supporting/contradictory literature:

- *Authors cite a broad range of previous zebrafish, mouse, and clinical studies linking ADAMTS13 to thrombosis, angiogenesis, and inflammation.*
- *Relevant negative or contradictory findings (e.g., studies suggesting ADAMTS13 is not essential outside haemostasis) could be discussed more explicitly to improve balance.*

Response:

We thank the reviewer for this valuable suggestion. Most prior studies of ADAMTS13 in animal models have centered on thrombotic thrombocytopenic purpura (TTP) and hemostasis or have examined protective, anti-inflammatory effects under stressful pathological conditions. These contexts could be interpreted as “negative” or “contradictory” in the sense that the literature provides little evidence for essential roles of ADAMTS13 outside hemostasis under basal conditions. By contrast, our work focuses specifically on developmental functions in the absence of inflammatory challenge, revealing immature or depleted macrophage populations and cytokine suppression and suggesting a primary developmental rather than stress-induced role. To explicitly address this point, we have revised the Discussion to highlight the predominance of hemostatic findings in the literature and to place our work in contrast.

Lines 228-235: Most prior studies of ADAMTS13 in animal models have focused on its role in hemostasis and TTP, or on its anti-inflammatory effects under stress or pathological conditions ... Our study provides new insights by revealing developmental functions of ADAMTS13 in angiogenesis, immune regulation, and hematopoiesis under baseline conditions.

Reviewer 2 - Comments and Author Response

1. ***First, authors validate the mutant allele by demonstrating the presence of schistocytosis, fragmentation of erythrocytes, in adult stages. This particular term should be defined the first time is being used.***

Response:

We thank the reviewer for this useful suggestion. We have now clarified the term in the following corrected sentence of the revised manuscript:

Lines 98-105: Flow cytometry analysis on peripheral blood from adult $a13^{i5}$ mutants revealed a shift toward smaller circulating cells, indicative of schistocytosis, the presence of fragmented erythrocytes with abnormal morphology (Kunitomo, Hirano and Tsuji, 2022)... consistent with the hemolytic features characteristic of cTTP and previously reported in zebrafish (Zheng et al., 2020).

2. ***Authors should define the casper allele and macrophage markers (marco and mpeg1) the first time they mention them in the ms.***

Response:

We agree with the reviewer’s comment. We have now included these definitions in the manuscript:

- *Line 55: Vascular Endothelial Growth Factor (VEGF).*
- *Line 125-128: to keep the terminology concise, the sentence now reads: To investigate the developmental pathophysiology ... and then introducing it into the transparent casper background (Figure 2B, C; see Methods for details on fish line generation).*
- *Line 133: days post-fertilization (dpf).*
- *Line 192: macrophage-specific markers (mpeg1 and marco).*

3. ***Sentence in line 46 of pg 10 has no-sense (“ADAMTS13 is required for defective hematopoietic lineage maturation”)***

Response:

We thank the reviewer for pointing out this error. The incorrect word “defective” (which was meant to read: *effective*) has now been removed. The revised sentence reads:

Lines 213-214: These findings suggest that ADAMTS13 is required for hematopoietic lineage maturation in the KM.

Second decision letter

MS ID#: bio.062265

MS Title: Zebrafish model reveals developmental and hematopoietic functions of ADAMTS13

Authors: Samuele Sartori, Ignacio Babiloni-Chust, Marco Varinelli, Alessandro Mattè, Piera Trionfini, Susanna Tomasoni and Lucia Poggi

I am happy to tell you that your manuscript has been accepted for publication in Biology Open, pending our standard publication integrity checks. It was accepted on 30th September 2025.